# Generalizing Adversarial Training to Composite Semantic Perturbations

**Yun-Yun Tsai** [1]   **Lei Hsiung** [1]   **Pin-Yu Chen** [2]   **Tsung-Yi Ho** [1]

## Abstract

Model robustness against adversarial examples has been widely studied, yet the lack of generalization to more realistic scenarios can be challenging. Specifically, recent works using adversarial training can successfully improve model robustness, but these works primarily consider adversarial threat models limited to $\ell_p$-norm bounded perturbations and might overlook semantic perturbations and their composition. In this paper, we firstly propose a novel method for generating composite adversarial examples. By utilizing component-wise PGD update and automatic attack- order scheduling, our method can find the optimal attack composition. We then propose **generalized adversarial training** (**GAT**) to extend model robustness from $\ell_p$ norm to composite semantic perturbations, such as Hue, Saturation, Brightness, Contrast, and Rotation. The results show that GAT can be robust not only on any single attack but also on combination of multiple attacks. GAT also outperforms baseline adversarial training approaches by a significant margin.

## 1. Introduction

Deep neural networks have shown remarkable success in a wide variety of real-life applications, ranging from biometric authentication (e.g., facial image recognition), medical diagnosis (e.g., CT lung cancer detection) to autonomous driving systems (traffic sign classification), etc. While these models can achieve great performance on benign data points, recent researches have shown models can be easily fooled by malicious data points crafted intentionally with adversarial perturbations, even with high standard accuracy (Szegedy et al., 2014).

[1]National Tsing Hua University, Hsinchu, Taiwan [2]IBM Research. Correspondence to: Yun-Yun Tsai <yt2781@columbia.edu>, Lei Hsiung <hsiung@m109.nthu.edu.tw>, Pin-Yu Chen <pinyu.chen@ibm.com>, Tsung-Yi Ho <tyho@cs.nthu.edu.tw>.

*Accepted by the ICML 2021 workshop on A Blessing in Disguise: The Prospects and Perils of Adversarial Machine Learning.* Copyright 2021 by the author(s).

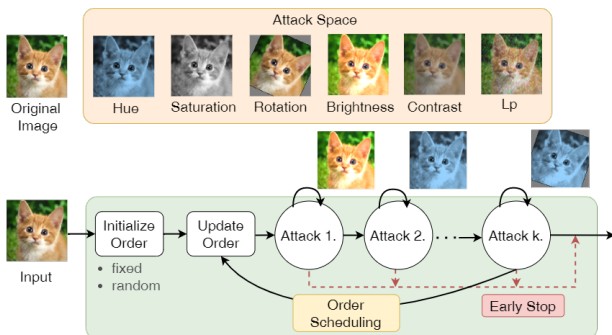

*Figure 1.* GAT implementation.

Therefore, the rapid growth of defense mechanisms attempts to improve models' robust accuracy against adversarial examples (Madry et al., 2018; Zhang et al., 2019). Nonetheless, most existing methods typically consider single threat models bounded by specific distance (e.g., $\ell_2$, $\ell_\infty$, etc.) and might overlook risk from the combination of multiple threats models. To tackle this issue, we propose a novel defense approach, named **generalized adversarial training** (**GAT**), which can harden against a wide range of threat models, from single $\ell_\infty$ and semantic (e.g., Hue, Saturate, Rotation, Brightness, Contrast) to the combination of them. Furthermore, we empirically discover adversarial training via combining each threat model sequentially in different orders has a significant influence on the robust accuracy of classifiers. As Fig. 1 shows the flow of GAT, we choose several attacks from attack pools, then sequentially combining the perturbations of them with a limited number of running steps. The order of combinations can be scheduled by the end of the combination in each round based on the order scheduling process. We propose a more potent composite threat model that allows scheduling the order from attack pools and optimizing perturbations of each attack component to evaluate the robustness of our GAT.

Different from existing works, this paper aims to address the following questions: (a) Can we generalize adversarial training from specific single threat models to multiple? (b) Can we optimize the order of combination between semantic and $\ell_p$ norm perturbations? (c) Can GAT outperform other adversarial training baselines for either $\ell_p$ or unseen threat models?

Our main contributions in this paper provide affirmative

answers to the aforementioned fundamental questions.

1. We propose **GAT**, a novel and unified approach to defense composite adversarial examples generated from multiple threat models, including $\ell_\infty$, Hue, Saturation, Rotation, Brightness, and Contrast. To the best of our knowledge, GAT is the first framework that robust on both single and composite adversarial attacks.

2. We elucidate the design principles of our component-wise projected gradient descent (PGD) for updating parameterized perturbations in each semantic attack. We also propose using order scheduling with the mirror descent algorithm (MDA) to optimize the scheduling matrix, which can further strengthen the composite threat models.

3. We evaluate robust accuracy for our GAT and compare it with other adversarial training benchmarks (Laidlaw et al., 2020; Madry et al., 2018; Zhang et al., 2019). Our results show that GAT outperforms them on all composite adversarial threats by $25\% \sim 40\%$ in random order, and by $8\% \sim 25\%$ in scheduled order, suggesting that GAT is an effective approach to defend multiple adversarial attacks.

## 2. Related Work

### 2.1. Adversarial Semantic Perturbations

Most recent studies on adversarial machine learning (ML) focus on generating examples that can fool the neural network to make the wrong prediction (Biggio & Roli, 2018). Several works have primarily focused on the vulnerability of deep neural networks against general $\ell_p$ adversarial threats (Goodfellow et al., 2015; Carlini & Wagner, 2017; Chen et al., 2018). Some others consider the adversarial threats beyond $\ell_p$, which generally occurs in natural transformation such as geometry, color, and lightness, named semantic perturbations. Typically, in contrast to the $\ell_p$ norm perturbations, which usually have a specific bound for perturbation, semantic attacks are not suitable for this restriction and parametrized them can be a challenging task. For generating semantic perturbations with color translation, (Hosseini & Poovendran, 2018) randomly choose the Hue value in HSV color space. (Laidlaw & Feizi, 2019) update 3-dimensional values in LUV space with PGD. For geometric transformation, (Xiao et al., 2018; Engstrom et al., 2019) target rotate transformation, the former used coordinate-wise optimization in each pixel, which is computationally expensive. The latter propose a simple way by parametrizing a set of tunable parameters for spatial transformation. Also, (Wong et al., 2019) defined Wasserstein distance for those adversarial examples with large norm bounded. In (Mao et al., 2020), they first propose using genetic algorithms for searching

the best combination of multiple adversarial attacks that are stronger than single adversarial attacks. However, they separately search the examples in subspace corresponding to the same norm.

### 2.2. Adversarial Training

The most well-known approach for learning a robust model is adversarial training (Madry et al., 2018; Zhang et al., 2019). However, most of them only target on the single adversarial threat. Specifically, a robust classifier that can help defend against a specific $\ell_p$ threat model still has low robustness to either $\ell_q$ threats ($p \neq q$) or semantic threats. The adversarial robustness under multiple adversarial threats has been discussed in (Tramèr & Boneh, 2019; Maini et al., 2020). They propose multiple norm adversarial training, which yield models simultaneously robust against multiple $\ell_p$ (e.g., $\ell_1$, $\ell_2$, and $\ell_\infty$) attacks. In contrast to their works, we shed the light on the significant of model robustness against multiple threats not only in $\ell_p$ but also semantic perturbations.

## 3. Generalized Adversarial Training

This section presents our proposed method, named GAT, for adversarial training with composite adversarial examples and an order scheduling algorithm for multiple attacks. A schematic overview of GAT is illustrated in Fig. 1.

### 3.1. Problem Formulation

**Composite adversarial attacks with order scheduling**: We propose method to generate adversarial examples with the combination of multiple attack algorithms. Let $\mathcal{F} : \mathcal{X} \to \mathbb{R}^d$ be an image classifier defined on $X \in \mathcal{X}$, where $d$ is the number of classes and $\mathcal{F}(X) \in \mathbb{R}^d$. Suppose there is an attack space $\Omega_A = \{A_1, A_2, ...A_K\}$ with corresponding epsilon space $\{\epsilon_1, \epsilon_2, ...\epsilon_K\}$, where $K$ is the number of attacks. An input $X$ can be transformed to $X_{adv}$ via updating perturbations $\delta_k$ in each attack operation $A_k \in \Omega_A$, where $k \leq K$. By selecting one attack $A_k$, $\delta_k$ can be optimized through maximizing the classification error (e.g., cross-entropy loss $L_{ce}$) within a boundary $\epsilon_k$:

$$\underset{A_k(X+\delta)}{\arg \max} \left\{ L_{ce}(\mathcal{F}(A_k(X+\delta), y)), \ s.t. \ \|\delta\| \leq \epsilon_k \right\} \quad (1)$$

To combine multiple attacks from $\Omega_A$ in a specific order, we can define the associated scheduling matrix $Z$ for $X_{adv}$. Suppose we wish to combine $N$ attacks, the scheduling matrix $Z$ is then a $N \times N$ square matrix. In our cases, $Z_{ij}$, $i, j \leq N$ is binary. $Z_{ij} = 1$ means the attack $A_j$ is launched at time $i$. By construction, $Z$ is a doubly stochastic matrix, $1^\top Z = 1$ and $Z1 = 1$, $1$ is all-one vector. Relax $Z$ to a binary matrix, $Z = [Z_1, ...Z_N]$. From time step 1 to N, we

define our composite adversarial images $X_{c-adv}$ as Eq. 2

$$X_{c-adv} = Z_N^\top A(Z_{N-1}^\top A...(Z_1^\top A(X; \epsilon_1)...; \epsilon_{N-1}); \epsilon_N) \quad (2)$$

In every time step $t \in [1, N]$, we launch an attack indexed by $Z_t$ and optimize the $\delta_t$ within the constraint:

$$\|Z_t^\top A(X_{t-1}; \delta_t) - Z_{t-1}^\top A(X_{t-2}; \delta_{t-1})\| \le \epsilon_t.$$

**Sinkhorn Operator**: Learning an optimal order which is discrete and non-differentiable from scheduling matrix can be viewed as a problem in differentiable relaxation. In (Mena et al., 2018), they theoretically showed how to extend Sinkhorn operator (Sinkhorn, 1966) to learn over the permutations. Similar as them, in our case, choosing an order over doubly stochastic matrix $Z$ can be cast as a maximization problem, where the set of $Z$ is convex belonging to the Birkhoff polytope $\mathcal{B}$. In more specific, we formulate our problem in Eq. 3. $P$ denotes the doubly stochastic matrix in the set $\mathcal{B}_P$, $\langle P, Z \rangle_\Phi = \sum_{i=1}^N (\max_i(P_i) \cdot Z_i)$.

$$\arg\max_{P \in \mathcal{B}_P} \langle P, Z \rangle_\Phi \quad (3)$$

Our goal aims to use $P$ to approximate the optimal $Z$. To update the scheduling matrix $Z$, we can use mirror descent algorithm (Wang & Banerjee). The update function can be formed as a double-loop algorithm

$$Z_{ij}^{t+\frac{1}{2}} = Z_{ij}^t exp(\frac{\partial L_{ce}(\cdot)}{\partial Z_{ij}})$$
$$Z_{ij}^t = \Pi_{\mathcal{B}_k}(Z_{ij}^{t+\frac{1}{2}}) \quad (4)$$

where $\Pi_{\mathcal{B}_k}$ is the projection function for $Z$ to project back onto the Birkhoff polytope which can be solved by Sinkhorn algorithm in a limit of iterations $i$ (Sinkhorn & Knopp, 1967).

### 3.2. Component-wise Projected Gradient Descent

For most of the semantic perturbations, their parameters are within the continuous value. In (Mohapatra et al., 2020), they show how semantic perturbations can be transformed from semantic space to general $\ell_p$ space. Motivated by this, we propose to update the parameters of semantic attacks by gradient descent algorithm with specific projection to each continuously semantic space. We show how to update the parameters in correspondence to five different semantic perturbations, including (i) hue, (ii) saturation, (iii) brightness, (iv) contrast, and (v) rotation. As Eq. 5, we extend the iterative method (Kurakin et al., 2017) to optimize our semantic perturbations, which is defined as:

$$x^{t+1} = clip_\epsilon(x^t + \alpha * sign(\nabla_{x^t} J(x^t, y))), \quad (5)$$

where $\alpha$ is a small step size, $J(\cdot)$ is the loss function (e.g., cross entropy), and we denote the $clip_\epsilon(z)$ as:

$$clip_\epsilon(z) = \begin{cases} \epsilon & \text{if } \epsilon < z, \\ z & \text{if } -\epsilon \le z \le \epsilon, \\ -\epsilon & \text{if } z < -\epsilon. \end{cases} \quad (6)$$

**Hue**: The scale of this space is represented as a color wheel, ranging from 0 to $2\pi$. We define $\epsilon_h \in [0, \pi]$ as the bound of $\delta^h$, which is the variation of hue value, *i.e.* $|\delta^h| \le \epsilon_h$. Let $x^h$ denote the hue value of the image $x$ in HSV space. The variance $\delta^h$ is initially chosen from uniform distribution $\mathcal{U}(-\epsilon_h, \epsilon_h)$, and the hue value $x^h$ can be updated iteratively by Eq. 7:

$$\delta_{t+1}^h = clip_{\epsilon_h}(\delta_t^h + \alpha * sign(\nabla_{\delta_t^h} J(\cdot))),$$
$$x_{t+1}^h = (x_t^h + \delta_{t+1}^h) \mod (2\pi). \quad (7)$$

**Saturation**: This space determines the colorfulness of the image. If the saturation value $x^s$ of image $x$ gets closer to zero, the color becomes more gray. We define saturation factor $\epsilon_s \in \mathbb{R}_+$ and $|\delta_s| \le \epsilon_s$. Same as the Hue function, we randomly initialize $\delta_s$ from $\mathcal{U}(0, \infty)$ and optimize $x^s$ iteratively by using Eq. 8:

$$\delta_{t+1}^s = clip_{\epsilon_s}(\delta_t^s + \alpha * sign(\nabla_{\delta_t^s} J(\cdot))),$$
$$x_{t+1}^s = \min(\max(0, x_t^s * \delta_{t+1}^s), 1). \quad (8)$$

**Brightness and Contrast**: Different from Hue and Saturation, this space determines the lightness of images. The transformation is directly applied on the pixel space instead of HSV space. Let $\delta^b$, $\delta^c$ denote the variation of the brightness value and contrast factor, subject to $|\delta^b| \le \epsilon_b \in [-1, 1]$ and $\|\delta^c\| \le \epsilon_c \in \mathbb{R}_+$. Let $x_t$ denote the attacked image of $x$ in step $t$. The iterative relation is defined as Eq. 9:

$$x_{t+1} = \min(\max(0, (x_t * \delta_{t+1}^c) + \delta_{t+1}^b), 1). \quad (9)$$

**Rotation**: For this transformation, we wish to find parameter $\theta$ such that rotating image around the center in an angle can be misclassified by model. Here, we define the rotate angle $\epsilon_\theta \in [-\pi, \pi]$ and $\delta_\theta \le \|\epsilon_\theta\|$. Given an input image $x$, the pixel index of $X$ is $[u, v]$, the center of $x$ is $[u_c, v_c]$, and the scale factor is $\gamma \in [0, 1]$. As Eq. 10 shows, we construct the transformed function for each index $[u, v]$. Here, $\mathcal{P} = \gamma \cos(\theta_t + \delta_{t+1}^\theta)$ and $\mathcal{R} = \gamma \sin(\theta_t + \delta_{t+1}^\theta)$.

$$\delta_{t+1}^\theta = clip_{\epsilon_\theta}(\delta_t^\theta + \alpha * sign(\nabla_{\delta_t^\theta} J(\cdot))),$$
$$\begin{bmatrix} u_{t+1} \\ v_{t+1} \end{bmatrix} = \begin{bmatrix} \mathcal{P}u_t & \mathcal{R}v_t & (1 - \mathcal{P}) \cdot u_c - \mathcal{R} \cdot v_c \\ -\mathcal{R}u_t & \mathcal{P}v_t & \mathcal{R} \cdot u_c + \mathcal{P} \cdot v_c \end{bmatrix}. \quad (10)$$

### 3.3. Generalized Adversarial Training (GAT)

To harden the classifier against multiple composite examples, we wish to extend the basic adversarial training approach. Suppose we wish to train a model $\mathcal{F}(\cdot)$ over a

| Training | Clean | Semantic attacks | | Full attacks | |
|---|---|---|---|---|---|
| | | Rand. | Sched. | Rand. | Sched. |
| Normal$^\dagger$ | 95.2 | 63.5 | 47.8 | 1.1 | 0.1 |
| Normal* | 94.0 | 53.5 | 34.2 | 0.8 | 0 |
| PAT$^\dagger_{self}$ | 82.4 | 41.4 | 22.5 | 27.7 | 11.8 |
| PAT$^\dagger_{alex}$ | 71.6 | 33.7 | 16.9 | 25.1 | 10 |
| Madry$^\dagger_{\ell_\infty}$ | 87.0 | 44.4 | 25.1 | 26 | 16.1 |
| Trades$^*_{\ell_\infty}$ | 84.9 | 37.2 | 11.5 | 13.4 | 7.6 |
| **GAT-s**$^*$ | 84.6 | **70.6** | **60.8** | **49.6** | **37.1** |
| **GAT-f**$^*$ | 84.7 | **72.7** | **64** | **53.2** | **45.5** |

*Table 1.* Comparison of performance on models using different adversarial training approaches. The notation † means the model is trained with Resnet50 and ∗ is WideResnet34. Additional detail of baseline models are described in Appendix 6.3

general data distribution $(x, y) \sim \mathcal{D}$ and also robustify against the adversarial data distribution $(x', y') \sim \mathcal{D}'$ generated from $(x, y)$. GAT follows the min-max optimization function in (Zhang et al., 2019). In particular, the inner maximization is solved using composite adversarial attacks proposed in Eq. 11

$$\min_{\mathcal{F}} \mathbb{E}_{(x,y)\sim\mathcal{D}} \left\{ \mathcal{L}(\mathcal{F}(x), Y) + \max_{x' \in \mathcal{B}(x, \{\epsilon\})} \mathcal{L}(\mathcal{F}(x'), \mathcal{F}(x))/\lambda \right\} \quad (11)$$

where $\lambda$ is a balance factor between natural loss and robust loss. The training procedure aims to minimize a regularized surrogate loss $\mathcal{L}(\cdot)$ by optimizing $\mathcal{F}$.

# 4. Experiments

In this section, we first elucidate the experimental settings and then present the performance evaluation of the proposed defense method GAT against multiple composite attacks on the Cifar-10 dataset (Krizhevsky, 2009).

**Attack and Training Setting**: To evaluate the robustness of the classifier, we generate the adversarial examples by using Cifar10 test sets, including 10000 images. For each attack $A_k \in \Omega_A$, we restrict $\epsilon_k$ within specific interval (in Appendix 6.1). We adopt the whole training sets, including 50000 images for doing generalized adversarial training and apply WideResNet34 (Zagoruyko & Komodakis, 2016) as the model architecture in two training settings, including training from scratch (**GAT-s**) and finetuning on the pretrained model (**GAT-f**) training with TRADES$_{\ell_\infty}$ (Zhang et al., 2019). Here, we report robust accuracy (RA), which is the percentage of inputs for which the model against the adversarial threats.

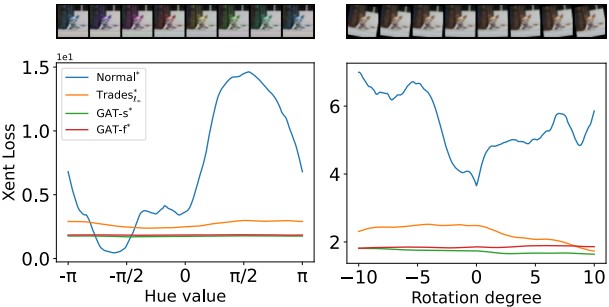

*Figure 2.* Loss landscape of random example when performing specific range of Hue ($\pm\pi$) and Rotations ($\pm10°$) in different adversarial training approaches.

## 4.1. Experimental Results

In Table 1, we show the RA (%) of composite semantic and full attacks. The composite semantic attack consists of the combination of 5 attacks. For instance, one can generate an example with *Hue, Saturation, Rotation, Brightness* and *Contrast* in a specific order. For full attacks, we generate examples with *All 5 semantic attacks* + $\ell_\infty$. We evaluate RA in two order settings for each attack: random and scheduled. The result shows GAT-s outperforms other baselines by 13% $\sim$ 49.3% on semantic attacks and 21% $\sim$ 37.1% on full attacks. More importantly, our GAT-f can further improve the RA by 3.2% $\sim$ 8.4 %, compared to GAT-s. Table **??** shows the RA of three composite attacks with different combinations and other results are shown in the Appendix 6.6. To better understanding the improvement in our approaches, we do the analysis of loss landscape. As Fig. 2 shows, the curve of cross entropy (Xent) loss of one random adversarial example becomes smoother and lower in our GAT. More visualization are shown in Appendix 6.2.

# 5. Conclusion

In this paper, we proposed GAT, a novel approach to strengthen the classifiers on those composite adversarial examples via using component-wise PGD and scheduling algorithm to find the worst-case examples from multiple semantic spaces. Compared to the existing adversarial training method, GAT allowed the classifier to defend against a variety of adversarial threats, from $L_p$ norms to semantic. Evaluated on the Cifar-10 dataset with two training settings for GAT, including training from scratch and finetuning on TRADES-$\ell_\infty$ pretrained model, our results demonstrated that GAT achieved high robust accuracy on most composite attacks, which provided a new perspective for defense method on multiple adversarial threats.

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

# 6. Appendix

## 6.1. Attack power settings

|  | Hue, $\epsilon_h$ | Saturate, $\epsilon_s$ | Bright, $\epsilon_b$ | Contrast, $\epsilon_c$ | Rotate, $\epsilon_\theta$ | $\ell_\infty, \epsilon_\ell$ |
|---|---|---|---|---|---|---|
| Range of Attack Power | $-\pi \sim \pi$ | $0.7 \sim 1.3$ | $-0.2 \sim 0.2$ | $0.7 \sim 1.3$ | $-10° \sim 10°$ | 8/255 |

*Table 2.*

## 6.2. Analysis of Loss Landscape and the Component-wise PGD Visualization

To better understanding why GAT leads to great improvement, in Fig. 3, we compare the curve of loss landscape for models trained by our GAT (from scratch and finetune) with others trained by TRADES($\ell_\infty$) and without adversarial training. We visualize the loss landscape of random example when performing different attacks in an inference time. We empirically observe that our models have more smoother and lower curve (red/green line) than the others on all of semantic single attacks, which shows the effectiveness of using composite adversarial examples for adversarial training.

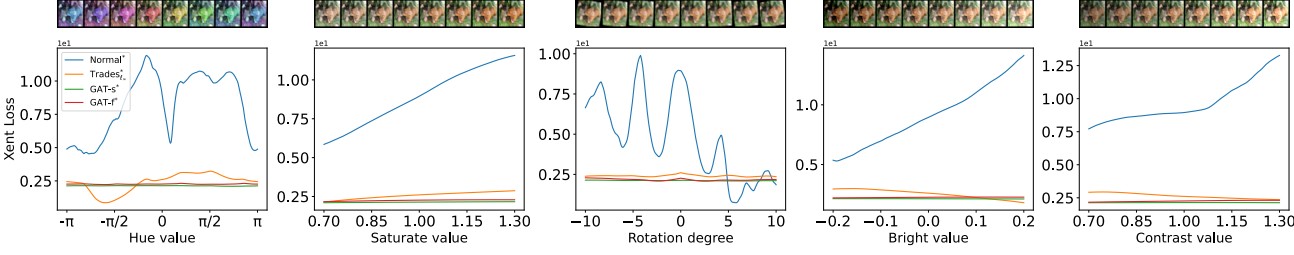

*Figure 3.* The loss landscapes of each single semantic attack with different WideResNet34 training models.

To show the effectiveness of component-wise PGD (comp-PGD), in Fig. 4, we simulate the update process of comp-PGD when performing semantic attack. For each attack, we propose to find the adversarial examples via sampling 20 random start points for updating $\epsilon$. We observe that in semantic attacks, comp-PGD indeed help searching the worst case by maximizing the loss during each attack.

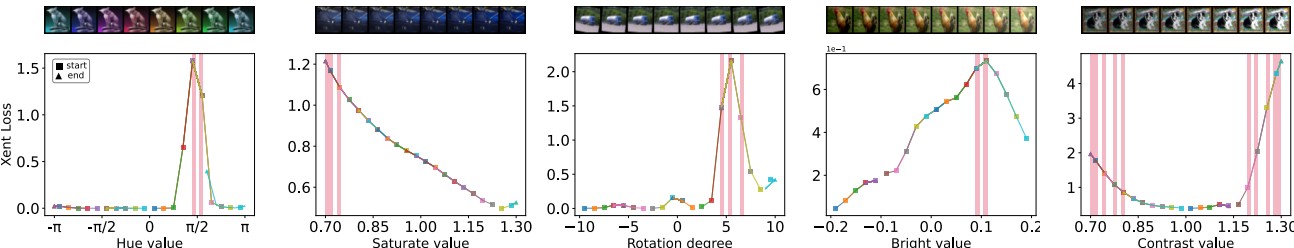

*Figure 4.* Component-wise PGD process of the single semantic attack. The red-marked area indicates where the semantic value, visited during iterations, will cause the natural model being misclassified. Each line segment means searching from *square* end to *triangle* end.

### 6.3. Model Details

Here, we show the details of baseline models training with different adversarial training approaches. The marker † means using the architecture ResNet50 for training and ∗ means using WideResNet34 for training.

**Normal**[†]: The model training with 50000 clean images on ResNet50.
**Normal**[∗]: The model training with 50000 clean images on WideResNet34.
**PAT**$_{self}^{\dagger}$: The pretrained model training with perceptual adversarial training, where the adversarial examples correlates with the LPIPS distance (based on ResNet50) from the natural example (Laidlaw et al., 2020).
**PAT**$_{alex}^{\dagger}$: The pretrained model training with perceptual adversarial training, where the adversarial examples correlates with the LPIPS distance (based on AlexNet) from the natural example (Laidlaw et al., 2020).
**Madry**$_{\infty}^{\dagger}$: The pretrained model training with $L_p$ adversarial training proposed in (Madry et al., 2018).
**Trades**$_{\infty}^{*}$: The pretrained model training with $L_p$ adversarial training proposed in (Zhang et al., 2019).

### 6.4. Implementation Details

For the attack part, we set the inner iteration number $i$ as 5, which means each component has 5 steps during PGD. Due to the non-convex feature of semantic attacks, in the process of each component doing PGD, we trigger the early-stopped condition to achieve the maximum attack success rate, once the adversarial example has been successfully generated. Notably, in the setting using scheduled order, we have an outer loop for updating the scheduling matrix with at most 5 times.

For the adversarial training part, we apply GAT with the WideResNet34 network (Zagoruyko & Komodakis, 2016). To make our training faster, we rather using composite examples update with random value in each semantic attack than using examples update with PGD step.

### 6.5. Algorithm Details

---

**Algorithm 1**

---

**Input:** classifier $f(\cdot)$, Attack space $\Omega_A$, input $x$, label $y$, step $\alpha$, maximum number of iterations $T$, number of gradient update iteration $K$, doubly stochastic matrix $Z$, step size $\{\alpha_t\}_{t=1}^{T}$
**Output:** Optimal adversarial examples $x'$

1: Choose $N$ attacks from $\Omega_A$; set $t = 1$
2: Initialize $Z$: Randomly permute an Identity matrix $I_{N \times N}$; Get initial order vector $S$.
3: Randomly initialize parameter set for N attacks $\{\epsilon_t\}_{t=1}^{N}$
4: $x' = x + 0.001 * \mathcal{N}(0, 1)$ ▷ Initialize perturbation with Gaussian noise
5: **while** $t \leq T$ **do**
6:  **for** $i = 1; i \leq N$ **do**
7:   ▷ Launch Attack $S_i^t$, $x_{temp}^0 \leftarrow x_i'$
8:   **while** $k \leq K$ **do**
9:    $x_{temp}^{k+1} = A_{S_i^t}\Big(Clip_{x,\epsilon_{S_i^t}}\big\{x_{temp}^k + \alpha * sign(\nabla_{x_{temp}^k} J(x_{temp}^k, y))\big\}\Big)$ ▷ PGD update for each attack by (5)
10:   **end while**
    $x_{i+1}' \leftarrow x_{temp}^K$
11:  **end for**
12:  $x_{adv} \leftarrow x_i'$ Evaluate $Loss$ in (1) with $X_{adv}$
13:  **if** $F(X_{adv} \neq y)$ **then**
14:   Break
15:  **else**
16:   ▷ Schedule Order with $Loss$ and $Z$ by (4)
17:   $Z^{t+\frac{1}{2}} = Z^t exp(\frac{\partial L_{ce}(\cdot)}{\partial Z_{ij}})$
18:   $Z_{ij}^{t+1} = \Pi_{\mathcal{B}_k}(Z_{ij}^{t+\frac{1}{2}})$
19:   $S^{t+1} \leftarrow \arg\max_i(Z_i)$
20:  **end if**
21: **end while**

---

## 6.6. Additional Experiment Results and Adversarial Examples

In Table 3 and Table 4, since the single attacks have no advantage in combination, the loss did not drop too much in this setting. We can see that for *normal* training models, the defense against semantic attacks is generally better than other adversarial training models. Also, in Table 9 and Table 10, GAT and normal training models perform well for semantic attacks. However, once the $\ell_\infty$ attack is enabled, the normal training models lost its advantages. In Table 5 and Table 6, we can see that although the *two attacks* do not outperforms, the GAT can be greatly improved in the *three attacks* (Table 7 and Table 8) and the *full attack* (Table 9 and Table 10) once the number of enabled attacks in the combination is increased.

### 6.6.1. SINGLE ATTACK

| Training | Clean | Single attack | | | | | |
| | | Hue | Saturate | Rotation | Brightness | Contrast | $\ell_\infty$ |
|---|---|---|---|---|---|---|---|
| Normal$^\dagger$ | 95.2 | 82.4 (12.8) | 94.6 (0.6) | 88.7 (6.5) | 93.3 (1.9) | 94.5 (0.7) | 1.9 (93.3) |
| Madry$^\dagger_{\ell_\infty}$ | 87.0 | 72.1 (14.9) | 86 (1) | 81.2 (5.8) | 81.4 (5.6) | 83.6 (3.4) | 73.7 (13.3) |
| PAT$^\dagger_{self}$ | 82.4 | 65.9 (16.5) | 81 (1.4) | 75.5 (6.9) | 77 (5.4) | 80.3 (2.1) | 65.8 (16.6) |
| PAT$^\dagger_{alex}$ | 71.6 | 55.4 (16.2) | 70.4 (1.2) | 65.1 (6.5) | 65.7 (5.9) | 68.5 (3.1) | 59.2 (12.4) |
| Normal$^*$ | 94.0 | 76.7 (17.3) | 93.2 (0.8) | 87.5 (6.5) | 91.1 (2.9) | 92.5 (1.5) | 1.5 (92.5) |
| Trades$^*_{\ell_\infty}$ | 84.9 | 67.3 (17.6) | 84 (0.9) | 78.9 (6) | 76.1 (8.8) | 77.7 (7.2) | 72.6 (12.3) |
| **GAT-s**$^*$ | 84.6 | 80.5 (4) | 83.5 (1) | 81.1 (3.4) | 82.3 (2.2) | 83.2 (1.4) | 63.9 (20.6) |
| **GAT-f**$^*$ | 84.7 | **83.2 (1.5)** | 83.8 (0.9) | **81.3 (3.4)** | **83.2 (1.5)** | 82.6 (2.1) | 68.4 (16.3) |

*Table 3.* Robust accuracy of single attack, which is one of semantic attacks, on Cifar-10.

| Training | Clean | Single attack | | | | | |
| | | Hue | Saturate | Rotation | Brightness | Contrast | $\ell_\infty$ |
|---|---|---|---|---|---|---|---|
| Normal$^\dagger$ | 0.0 | 13.8 | 0.8 | 7.5 | 2.2 | 0.9 | 98.0 |
| Madry$^\dagger_{\ell_\infty}$ | 0.0 | 17.7 | 1.4 | 7.4 | 7.1 | 4.3 | 15.3 |
| PAT$^\dagger_{self}$ | 0.0 | 21.3 | 2.0 | 10.1 | 7.5 | 3.1 | 20.1 |
| PAT$^\dagger_{alex}$ | 0.0 | 25.1 | 2.4 | 11.4 | 10.6 | 6.0 | 17.3 |
| Normal$^*$ | 0.0 | 19.0 | 1.0 | 7.5 | 3.3 | 1.8 | 98.4 |
| Trades$^*_{\ell_\infty}$ | 0.0 | 21.2 | 1.3 | 8.0 | 11.3 | 9.2 | 14.5 |
| **GAT-s**$^*$ | 0.0 | 5.2 | 1.4 | 4.7 | 2.9 | 2.0 | 24.4 |
| **GAT-f**$^*$ | 0.0 | **1.9** | 1.3 | **4.5** | **2.0** | 2.7 | 19.2 |

*Table 4.* Attack success rate of single attack.

### 6.6.2. TWO ATTACKS

| Training | Clean | 2 attacks ($\ell_\infty$ + *one of semantic attacks*) | | | | |
| | | Hue | Saturate | Rotation | Brightness | Contrast |
|---|---|---|---|---|---|---|
| Normal$^\dagger$ | 95.2 | 2.1 (93.1) | 1.8 (93.4) | 2.3 (92.9) | 2 (93.2) | 2.2 (93) |
| Madry$^\dagger_{\ell_\infty}$ | 87.0 | 61.4 (25.6) | 72.3 (14.7) | 67.6 (19.4) | 68 (19) | 70.2 (16.8) |
| PAT$^\dagger_{self}$ | 82.4 | 48 (34.4) | 64.1 (18.3) | 56.6 (25.6) | 59.8 (22.6) | 63.1 (19.3) |
| PAT$^\dagger_{alex}$ | 71.6 | 43.1 (28.5) | 57.7 (13.9) | 51.6 (20) | 52.8 (18.8) | 55.5 (16.1) |
| Normal$^*$ | 94.0 | 1.1 (92.9) | 1.4 (92.6) | 1.8 (92.2) | 1.3 (92.7) | 1.6 (92.4) |
| Trades$^*_{\ell_\infty}$ | 84.9 | 59.7 (25.2) | 71.1 (13.8) | 65.9 (19) | 64.5 (20.4) | 65.6 (19.3) |
| **GAT-s**$^*$ | 84.6 | 60.2 (24.4) | 62.9 (21.7) | 60.6 (24) | 62.1 (22.5) | 61.3 (23.3) |
| **GAT-f**$^*$ | 84.7 | **67.2 (17.5)** | 66.9 (17.8) | 64.6 (20.1) | 66.9 (17.8) | 65.6 (19.1) |

*Table 5.* Robust accuracy of two attacks.

| Training | Clean | 2 attacks ($\ell_\infty$ + *one of semantic attacks*) | | | | |
|---|---|---|---|---|---|---|
| | | Hue | Saturate | Rotation | Brightness | Contrast |
| Normal[†] | 0.0 | 97.8 | 98.2 | 97.5 | 97.9 | 97.7 |
| Madry$_{\ell_\infty}^{\dagger}$ | 0.0 | 29.5 | 17.0 | 22.4 | 21.9 | 19.4 |
| PAT$_{self}^{\dagger}$ | 0.0 | 41.9 | 22.3 | 31.2 | 27.6 | 23.5 |
| PAT$_{alex}^{\dagger}$ | 0.0 | 40.5 | 19.5 | 28.4 | 26.9 | 22.7 |
| Normal* | 0.0 | 98.8 | 98.5 | 98.1 | 98.6 | 98.3 |
| Trades$_{\ell_\infty}^{*}$ | 0.0 | 29.7 | 16.3 | 22.4 | 24.0 | 22.8 |
| **GAT-s***  | 0.0 | 28.8 | 25.7 | 28.3 | 26.6 | 27.4 |
| **GAT-f***  | 0.0 | **20.6** | 21.0 | 23.7 | **21.0** | 22.5 |

*Table 6.* Attack success rate of two attacks.

### 6.6.3. MULTIPLE ATTACKS: 3 ATTACKS

Since the width of the page is limited, here we noted the attack in the following abbreviations:

- **0**: Hue, **1**: Saturate, **2**: Rotation, **3**: Brightness, **4**: Contrast, **5**: $\ell_\infty$.

- **s**: scheduled order, **r**: random order.

We enabled the $\ell_\infty$ attack in all *three attacks* to mix the norm spaces, which is semantic space, spatial space, and $L_p space$.

| Training | Clean | 3 attacks ($\ell_\infty$ + *two of semantic attacks*) | | | | | |
|---|---|---|---|---|---|---|---|
| | | (0,1,5) (r) | (0,1,5) (s) | (0,2,5) (r) | (0,2,5) (s) | (3,4,5) (r) | (3,4,5) (s) |
| Normal[†] | 95.2 | 1.7 (93.5) | 0.5 (94.7) | 1.9 (93.4) | 0.9 (94.3) | 1.8 (93.4) | 0.1 (95.1) |
| Madry$_{\ell_\infty}^{\dagger}$ | 87.0 | 52.1 (34.9) | 52.1 (34.9) | 55.7 (31.3) | 47.2 (39.8) | 47.2 (39.8) | 43.8 (43.2) |
| PAT$_{self}^{\dagger}$ | 82.4 | 47.1 (35.3) | 40.5 (41.9) | 55.6 (26.8) | 40.9 (41.5) | 37.8 (44.6) | 30.9 (51.5) |
| PAT$_{alex}^{\dagger}$ | 71.6 | 42.1 (29.5) | 35.2 (36.4) | 47.8 (23.8) | 34.4 (37.2) | 34.8 (36.9) | 27 (44.6) |
| Normal* | 94.0 | 1.1 (92.9) | 0.4 (93.7) | 1.3 (92.7) | 0.6 (93.4) | 1.1 (92.9) | 0.1 (93.9) |
| Trades$_{\ell_\infty}^{*}$ | 84.9 | 48.7 (36.2) | 49 (35.9) | 40.9 (44) | 28.8 (56.1) | 45.5 (39.4) | 42.6 (42.3) |
| **GAT-s***  | 84.6 | 59.3 (25.3) | 56.6 (27.9) | 58.9 (25.6) | 51.4 (33.1) | 56.1 (28.5) | 53 (31.6) |
| **GAT-f***  | 84.7 | **66.9 (17.8)** | **64.2 (20.5)** | **63.8 (20.9)** | **56.9 (27.8)** | **62.8 (21.9)** | **61.3 (23.4)** |

*Table 7.* Robust accuracy of three composite attacks.

| Training | Clean | 3 attacks ($\ell_\infty$ + *two of semantic attacks*) | | | | | |
|---|---|---|---|---|---|---|---|
| | | (0,1,5) (r) | (0,1,5) (s) | (0,2,5) (r) | (0,2,5) (s) | (3,4,5) (r) | (3,4,5) (s) |
| Normal[†] | 0.0 | 98.2 | 99.5 | 98.1 | 99.1 | 98.1 | 99.9 |
| Madry$_{\ell_\infty}^{\dagger}$ | 0.0 | 40.2 | 40.1 | 36.0 | 45.8 | 45.8 | 49.7 |
| PAT$_{self}^{\dagger}$ | 0.0 | 42.9 | 50.8 | 32.6 | 50.3 | 54.2 | 62.5 |
| PAT$_{alex}^{\dagger}$ | 0.0 | 41.6 | 50.8 | 33.7 | 52.0 | 51.8 | 62.3 |
| Normal* | 0.0 | 98.8 | 99.6 | 98.6 | 99.4 | 98.8 | 99.9 |
| Trades$_{\ell_\infty}^{*}$ | 0.0 | 42.7 | 42.3 | 51.9 | 66.1 | 46.4 | 49.9 |
| **GAT-s***  | 0.0 | 29.9 | 33.0 | 30.3 | 39.2 | 33.7 | 37.3 |
| **GAT-f***  | 0.0 | **21.0** | **24.1** | **24.6** | **32.8** | **25.8** | **27.6** |

*Table 8.* Attack success rate of three composite attacks.

### 6.6.4. MULTIPLE ATTACKS

| Training | Clean | Semantic attacks | | Full attacks | |
|---|---|---|---|---|---|
| | | Random | Scheduled | Random | Scheduled |
| Normal[†] | 95.2 | 63.5 (31.7) | 47.8 (47.4) | 1.1 (94.1) | 0.1 (95.1) |
| Madry$_{\ell_\infty}$[†] | 87.0 | 44.4 (42.6) | 25.1 (61.9) | 26 (61) | 16.1 (70.9) |
| PAT$_{self}$[†] | 82.4 | 41.4 (41) | 22.5 (59.9) | 27.7 (54.7) | 11.8 (70.6) |
| PAT$_{alex}$[†] | 71.6 | 33.7 (37.9) | 16.9 (54.7) | 25.1 (46.5) | 10 (61.6) |
| Normal* | 94.0 | 53.5 (40.5) | 34.2 (59.8) | 0.8 (93.2) | 0 (94) |
| Trades$_{\ell_\infty}$* | 84.9 | 37.2 (47.7) | 11.5 (73.4) | 13.4 (71.5) | 7.6 (77.3) |
| **GAT-s*** | 84.6 | 70.6 (14) | 60.8 (23.7) | 49.6 (35) | 37.1 (47.4) |
| **GAT-f*** | 84.7 | **72.7 (12)** | **64 (20.7)** | **53.2 (31.5)** | **45.5 (39.2)** |

*Table 9.* Robust accuracy of composite semantic attacks and composite full attacks.

| Training | Clean | Semantic attacks | | Full attacks | |
|---|---|---|---|---|---|
| | | Random | Scheduled | Random | Scheduled |
| Normal[†] | 0.0 | 33.4 | 49.8 | 98.8 | 99.9 |
| Madry$_{\ell_\infty}$[†] | 0.0 | 49.2 | 71.2 | 70.2 | 81.5 |
| PAT$_{self}$[†] | 0.0 | 50.0 | 72.7 | 66.4 | 85.6 |
| PAT$_{alex}$[†] | 0.0 | 53.5 | 76.4 | 64.9 | 86.0 |
| Normal* | 0.0 | 43.1 | 63.6 | 99.2 | 100.0 |
| Trades$_{\ell_\infty}$* | 0.0 | 56.2 | 86.4 | 84.2 | 91.0 |
| **GAT-s*** | 0.0 | 16.6 | 28.1 | 41.4 | 56.1 |
| **GAT-f*** | 0.0 | **14.2** | **24.4** | **37.1** | **46.2** |

*Table 10.* Attack success rate of composite semantic attacks and composite full attacks.

## 6.7. Additional Adversarial Examples

Here we provide some of the adversarial examples in above experimental settings. We arrange the images into different columns; the most left column is the original images; the following two columns are the examples and their magnified differences compared with the originals.

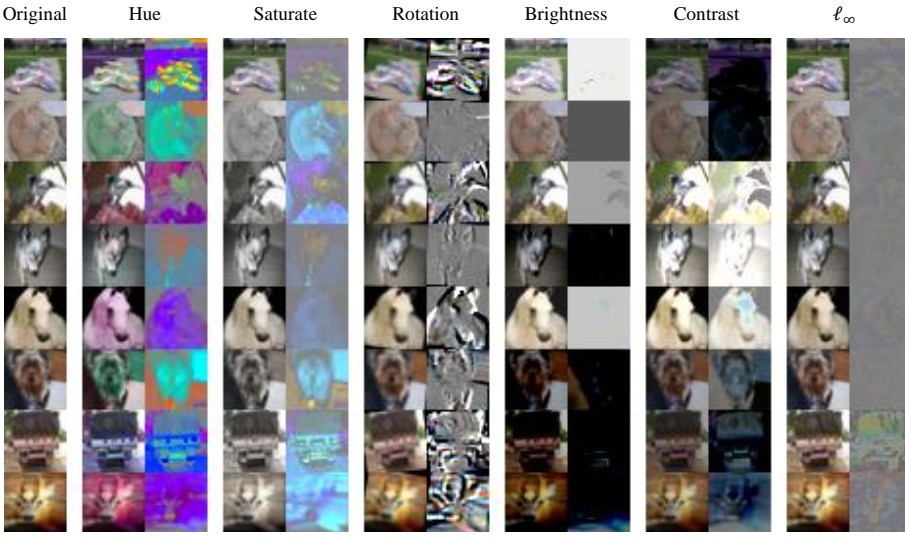

| Original | Hue | Saturate | Rotation | Brightness | Contrast | $\ell_\infty$ |

*Figure 5.* Adversarial examples generated under one of semantic attacks (hue, saturate, rotation, bright, contrast) or $\ell_\infty$ attack.

| Original | $\ell_\infty$, Hue | $\ell_\infty$, Saturate | $\ell_\infty$, Rotation | $\ell_\infty$, Brightness | $\ell_\infty$, Contrast |
|---|---|---|---|---|---|

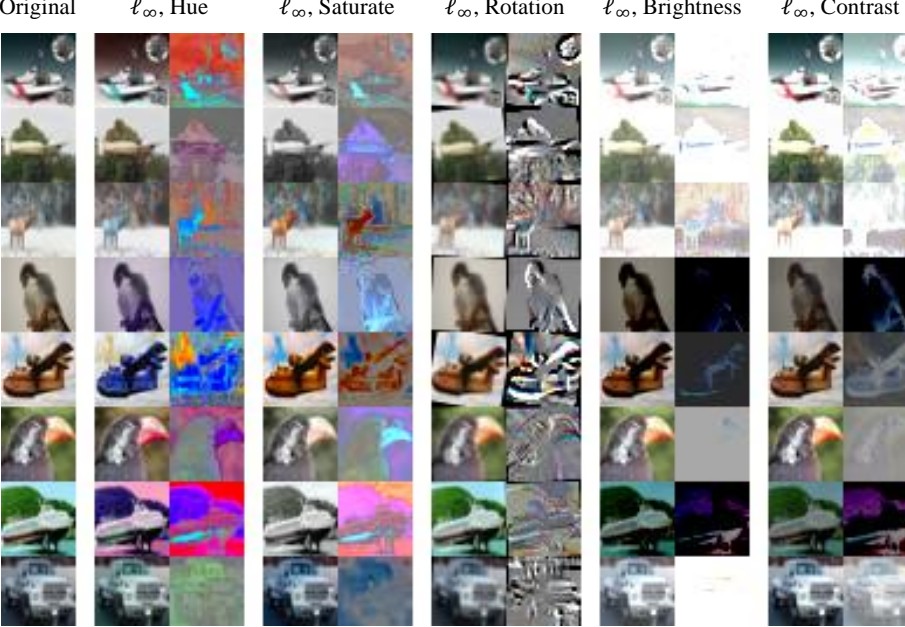

*Figure 6.* Adversarial examples generated under two attacks (composed of one semantic attack and the $\ell_\infty$ attack).

| Original | $\ell_\infty$, Hue, Sat. | $\ell_\infty$, Bri., Con. | $\ell_\infty$, Hue, Rot. | Semantic | Full |
|---|---|---|---|---|---|

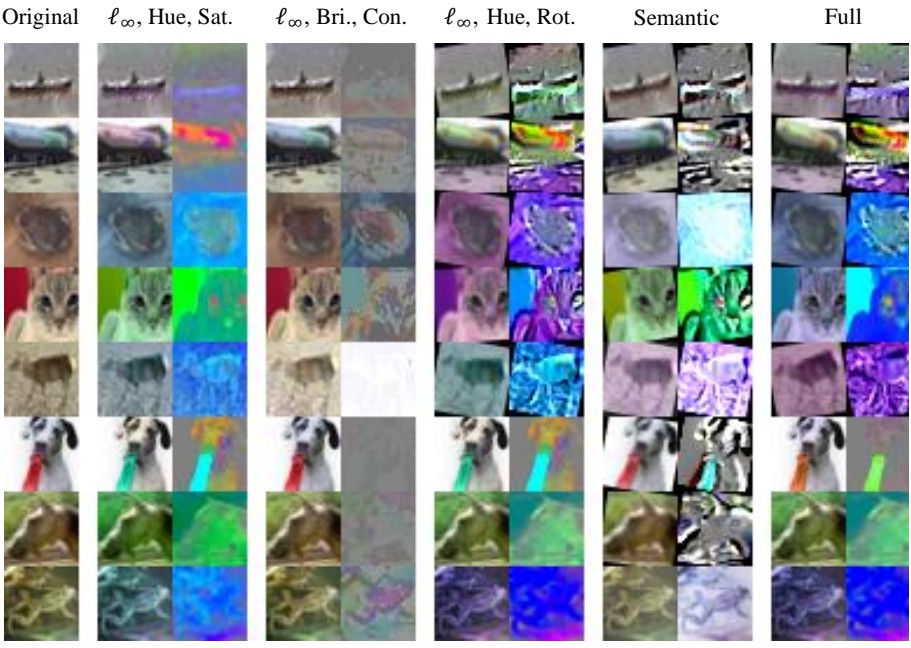

*Figure 7.* Adversarial examples generated under three and more attacks. *Semantic* means we enable all semantic attacks (hue, saturate, rotation, brightness, and contrast). *Full* means we enable $\ell_\infty$ and all semantic attacks.