# OpenReview forum: "Generalizing Adversarial Training to Composite Semantic Perturbations"
_ICML.cc/2021/Workshop/AML — ICML 2021 Workshop AML Poster_

### Official Review · Reviewer_epn2 · 2021-06-20
**It is a novel work in adversarial training which uses the worst-case samples from the combination of multiple semantic attacks**

**Rating:** Accept
**Confidence:** 4

**Review:**

This paper is well organized and the expression is clear.

This paper proposes a novel idea in adversarial training that adversarial samples are generated by both multiple semantic transformations and an $l_\infty$ threat model. This paper not only optimizes each attack models but also optimizes the order of them. This method significantly improves robust accuracy on attack models compared to sota adversarial training methods.

The experiments are solid in the comparison with existing adversarial training methods, while they are only conducted on CIFAR-10, which might weaken the effectiveness of the conclusions.

I’m confused by the notation $A$ in equation (2). Is $A$ a list composed by $N$ attacks selected from $\Omega_A$? This is necessary to be mentioned in your paper.

---

### Decision · Program_Chairs · 2021-06-21

**Decision:**

Accept (Poster)

**Comment:**

This paper proposed a novel idea in adversarial training. The experiments are solid.